# Histiocytic Sarcoma Involving Cervical Vertebra: A Case Report and Review of the Literature

**DOI:** 10.3390/brainsci12070958

**Published:** 2022-07-21

**Authors:** Eshagh Bahrami, Masoumeh Najafi, Amin Jahanbakhshi, Jaber Hatam, Saadat Molanaei, Patrizia Ciammella, Salvatore Cozzi

**Affiliations:** 1Skull Base Research Center, Department of Neurosurgery, Hazrat Rasoul Akram Hospital, Iran University of Medical Sciences, Tehran 1997667665, Iran; bahrami_eses@yahoo.com (E.B.); najafi.mas@iums.ac.ir (M.N.); jaber.hatam@yahoo.com (J.H.); 2Stem Cell and Regenerative Medicine Research Center, Iran University of Medical Sciences, Tehran 1997667665, Iran; 3Department of Pathology, Milad Hospital, Hakim Highway, Tehran 1997667665, Iran; saadatmolanaei@yahoo.com; 4Radiation Oncology Unit, Azienda USL-IRCCS di Reggio Emilia, 42123 Reggio Emilia, Italy; patrizia.ciammella@asmn.re.it (P.C.); salvatore.cozzi@hotmail.it (S.C.)

**Keywords:** histiocytic sarcoma, histocyte, radiotherapy, immunohistochemistry, vertebral column

## Abstract

Histiocytic sarcoma (HS) is a rare neoplasm composed of cells with immunohistochemical characteristics of mature histiocytes. It can be disseminated or localized and usually involves the skin, spleen, and gastrointestinal tract. Primary involvement of the vertebral column is extremely rare. We report a 29-year-old female who presented with neck pain and had a destructive 35*43*48 mm lesion in C2 with a paravertebral extension. The initial biopsy did not lead to the correct diagnosis. She later developed dysphagia, and the anterior approach was used for tumor decompression. The diagnosis of cervical histiocytic sarcoma was made, and she underwent radiotherapy. The follow-up MRI showed a marked response to radiotherapy. Here, we report the first case of cervical HS, review all cases of vertebral HS, compare patients’ characteristics and clinical courses, and discuss diagnostic nuances and treatment options.

## 1. Introduction

Histiocytic sarcoma (HS) is a malignant proliferation of cells showing the morphological and immunophenotypic characteristics of mature histiocytes. Histiocytes are phagocytic, antigen-presenting cells that originate from hematopoietic stem cells that perform the processing and destruction of antigens. Derived from these cells, HS is an aggressive neoplasm with poor outcomes. It can be disseminated or localized in the skin, lymph nodes, gastrointestinal tract, or other organs. HS was not a known lesion until the development of immunohistochemistry and the availability of molecular genetic tools, and most cases were previously diagnosed as lymphomas such as diffuse large B-cell lymphoma (DLBCL) or anaplastic large cell lymphoma (ALCL) [1,2]. The disease can appear as a primary malignancy or, usually, as a secondary malignancy. There are several reports hypothesizing the differentiation from DLBCL to HS, which accounts for about a quarter of cases [3]. HS has also been reported as a common secondary malignancy after gonadal or mediastinal germ cell tumors [4].

Primary HS in the spine is extremely rare and has been reported in only six cases [2,5,6,7,8,9]. The involvement of the cervical spine has not been reported until now. Because of this rarity, it is a diagnostic challenge, and there is no consensus on treatment. Here, we report the first case of primary HS in the C2 vertebra that showed an excellent response to radiotherapy. We discuss all the reported instances of vertebral HS in terms of clinical manifestations, immunohistochemical (IHC) findings, and treatment options. 

## 2. Case Presentation 

The patient was a 29-year-old female with an initial complaint of neck pain and limited mobility. In a cervical CT scan, the complete destruction of the second cervical vertebra was obvious. In the MRI of the cervical spine, a large heterogenous tumoral mass in the C2 vertebra with some necrotic/cystic parts and a marked paravertebral extension was seen. It was almost isointense to the cord in the T1W and T2W images and was heterogeneously enhanced with gadolinium (Figure 1). The thoracic and lumbar MRI and the thoracic and abdominal CT scans were normal. 

Because of vertebral instability, the patients underwent craniocervical fixation and tumor sampling through the C2 pedicle. The histopathologic examination reported an atypical histiocytic tumor infiltrated with hemophagocytosis. In the immunohistochemical examination, CD3, CD15, CD20, CD30, CD34, S100, GFAP, desmin, and EMA were negative and CD68, CD163, and vimentin were positive. ki67 was 15%. The positron emission tomography (PET) scan showed two FDG avid skeletal lesions, one in the C2 odontoid process (SUV max = 7.8) most consistent with Langerhans cell histiocytosis and the other one in the right iliac bone (SUV max = 4.5) consistent with the place of the bone graft extraction at surgery.

Three months later, the patient’s symptoms worsened, and she developed dysphagia. A repeat MRI showed a significant increase in the size of the tumor and a paravertebral extension (35 × 43 × 48 versus 52 × 42 × 60) (Figure 2). This time, an anterior approach was chosen for tumor decompression. There was cellular infiltration of the stroma characterized by sheets of large cells with abundant cytoplasm and rather distinct cell borders. Highly pleomorphic and sometimes hyperchromatic single nuclei were seen. Within the cytoplasm of most of the tumor cells, there were abundant partly degenerated RBCs. Scattered lymphoid cells were also present in the stroma. Some of the particles were bordered by tiny bony trabeculae. No necrosis was seen. Eight to ten mitoses per high power field with few atypical mitoses were detected, and the diagnosis of malignant HS was made. In the immunohistochemistry, cells were positive for CD45, CD43, CD68, and S100 (S100 negativity in the first operation may be due to the small size of the sample) and negative for CD3, CD20, and cytokeratin (Figure 3). The metastasis workups (thoracic and abdominopelvic CT scan) and bone marrow biopsy were normal. To deal with this new reported pathology, 25 sessions of radiotherapy (IMRT) in 5 weeks were considered for the patient to a total dose of 45Gy. 

At the end of the radiotherapy period, the patient suffered from a sudden monoparesis of the left upper limb. The CT scan indicated a hypodense lesion in the head of the caudate nucleus, putamen, and anterior thalamus, which was hyperintense in the T1- and T2-weighted MRI. In the MRA, the right internal carotid artery was not visible (Figure 4 and Figure 5). The patient also complained of dyspnea. The patient then underwent a cardiac MRI, which showed multiple masses in the left atrium spreading to the right upper pulmonary vein without obstruction. The patient had a history of open-heart surgery 5 years ago due to a heart mass, which was reported to be fibroma. According to the MRI and history, these lesions have been reported in favor of fibroma.

Six months after radiotherapy, the MRI showed a very good response to the treatment, with no mass lesion and non-specific enhancement at C2 and C3 (Figure 5). The thoracic and lumbar MRI and the thoracic and abdominopelvic contrast-enhanced CT scans were all normal. Three months later, she developed headaches and visual complaints. The MRI depicted a large right parietooccipital lesion with ring enhancement in favor of metastasis (Figure 6). We suggested surgery and radiation for the brain lesion and chemotherapy, but the patient refused any other treatment and therefore was a candidate for supportive care.

## 3. Discussion 

HS is a very rare neoplasm of the hematopoietic system that usually presents in extranodal sites as a painless mass. Common locations are the gastrointestinal tract, skin, spleen, and liver. The median age at diagnosis is 51, and it predominates in males [10]. It involves other sites much less frequently. The involvement of the central nervous system has been reported in 30 cases [11]. The primary involvement of the vertebral column by HS is extremely rare. There are very few cases reported in the literature [2,5,6,7], and here, we report the first case of primary HS in the cervical spine. 

The diagnosis of HS is based on histologic appearance and, more importantly, immunohistochemical tests. Langerhans cell sarcoma (LCS), peripheral T-cell lymphoma, diffuse large B-cell lymphoma, anaplastic large cell lymphoma (ALCL), and melanoma can make diagnosis a challenge. CD163 is very useful, is specific for HC, and was the only factor in Pileri’s series that was positive in all cases of HS [8]. Immunohistochemistry plays a crucial role in the diagnosis of HS. The IHC in our patient was negative for CD3, CD15, CD20, CD30, CD34, GFAP, desmin, EMA, and cytokeratin and positive for CD68, CD163 and vimentin, CD45, CD43, and CD68. S100 is a marker for melanoma and dendritic cell sarcoma. However, it may be positive in 33% of HSs as well [11]; as it was positive in the IHC after the second operation. Negativity for EMA and GFAP rules out the diagnosis of meningioma and glial-derived tumors. CD45 is positive in diffuse large B-cell lymphoma and anaplastic large cell lymphoma; but negative immunoreactivity for CD20, CD3, CD15, and CD30 excludes lymphomas. Negativity for CD34 is against the diagnosis of myeloid sarcoma and positivity for CD68 and CD163 is in favor of a histiocytic origin [12,13]. BRAF Val600Glu is a common mutation among histiocytic and dendritic cell tumors, including some HS cases [14], and there are reports that consider an important role for IGH rearrangement, specially in cases developing in the background of B cell lymphoma [15]. Ki67, a marker of cell proliferation, was 15% in our case, which is in the lower part of the reported spectrum in HS, which ranges from 10% to 90% [11]. This low proliferative index may show a less aggressive tumor behavior and might underlie a good response to treatment.

Except for two cases reported in 3- and 63-year-old patients, most cases of vertebral HS had involved young adults between 17 to 29 years old (Table 1). Treatment strategies varied from a combination of surgery and radiotherapy to using either modality alone. Most cases shared back pain and weakness as the presenting symptoms.

Pileri et al. published a series of patients with histiocytic tumors and reported the first available case of primary vertebral involvement, a 26-year-old male who presented with lytic lesions at the T4 and T5 vertebra. He was previously treated for L2 leukemia [8]. Lin et al. reported a 27-year-old woman with a 2-month history of radicular low back pain and mild motor weakness. L3 vertebra was involved with an anterior epidural mass. They treated the patient with complete L3 spondylectomy, and no adjuvant treatment was performed. The patient was disease-free after 2 years [7]. Kaushal reported a 17-year-old male with low back pain and left lower extremity weakness developed in 2 months. There was a partial involvement of the left part of L3 and epidural compression by the tumor. The patient was treated by surgery, followed by radiotherapy, and he was disease-free after 3.5 years [6]. Patnaik et al. introduced a 22-year-old male with a three-month history of weakness that progressed to paraplegia. There was sphincter dysfunction and a sensory level at T10. As in other cases, there was no other organ involvement. In the MRI, there was a tumor involving T8, T9, and partially T10 vertebra. He underwent surgical resection but refused to continue adjuvant treatments. He was alive 5 months after the procedure [2].

Buonocore et al. reported a 3-year-old boy who presented with low back pain. There were complete T6 and L4 collapses, with an epidural mass at L4. There was no other involved focus. They started chemotherapy (prednisone, 6-mercaptopurine, methotrexate, vinblastine, and etoposide, and after radiation, three monthly cycles of cyclophosphamide and actinomycin D) and radiotherapy (4500 cGy in 180-cGy fractions) to the L4 vertebra, which led to size reduction at the L4 epidural space. He then developed two pulmonary nodules which were managed again by chemotherapy (two monthly cycles of idarubicin and 2-chlorodeoxyadenosine-2CDA, and after radiotherapy, six monthly cycles of 2CdA) and radiotherapy (4500 cGy at 180 cGy per fraction). They found a poor response to LCH- and sarcoma-based chemotherapy and a slightly better response to idarubicin and 2CdA. They have noted that the clinical improvement was seen after the second half of the radiotherapy course, and, interestingly, the T6 lesion did not progress despite no radiotherapy at this level [5]. Sohn et al. reported a 63-year-old man with weight loss, fever, and general weakness. He had multiple vertebral bone marrow involvements in the lower thoracic and lumbar spine without a mass effect or vertebral destruction. The patient refused treatment and died 2 months later, and a diagnosis was made based on necropsy [9].

Getting all these reports together (Table 1), it is clear that vertebral HS is more common in males, which is concordant with previous reports [8]. Ma et al. reviewed primary central nervous system HS and revealed no sex predilection and a mean age of 43 years [11]. It seems that vertebral HS occurs in younger ages (median age: 26 years) compared to other types of HS (median age: 46 in Pileri et al. [8]). It is not possible to conclude a survival estimate because of the lack of sufficient evidence, but most of the reported vertebral HS had been disease-free for 2–3 years. All reported tumors were in the mid-thoracic or mid-lumbar spine, except our case, which was in the cervical region. The presenting symptoms were back spinal pain and a neurologic deficit, except for Sohn et al., who presented a disseminated vertebral disease.

There is no standard treatment for HS, but the most used treatments are a combination of surgery, radiotherapy, and chemotherapy. The optimal dose of radiotherapy and the chemotherapy regime is also unknown [14,16]. In the reported cases,

HS is a very aggressive tumor with a poor prognosis. According to a study by the National Cancer Database (NCDB) on 330 cases of HS, after a median follow-up of 16 months, 227 patients (69%) had died. The median OS was 6 (range, 1–127) months [16]. The data from the reported cases of vertebral HS (summarized in Table 1) do not give us enough information to calculate the overall survival. It seems that primary vertebral HS shows a good response to radiotherapy. In our case, treatment with the partial decompressive resection of the tumor and radiotherapy without chemotherapy led to a decent response, with near-complete tumor regression in the follow-up MRI. Therefore, we decided to postpone chemotherapy as a salvage therapy for the probable future recurrence of the tumor; however, after 9 months, when a brain metastasis was detected, she refused to be treated.

Recent advances in developing target therapies have shown some promise in the treatment of HS. Because of the involvement of the MAPK/ERK pathway, different molecules in this pathway, such as BRAF, MAPK, KRAS, and MEK, may be potential targets for the modification of the pathology. The MEK inhibitor trametinib and the BRAF inhibitor vemurafenib have been tried for histiocytic tumors, with acceptable responses in some cases [17].

## 4. Conclusions

Primary HS of the spinal column is extremely rare. One should have a strong suspicion and perform extensive IHC tests to detect the pathology. The treatment options are not standard, but most authors insist on surgery plus chemotherapy and radiotherapy. Although the surgical procedure attempted just a tumor debulking, this patient showed a very good radiologic response to radiotherapy. So, we showed that when a total resection of the tumor is not possible, radiotherapy with 4500 cGy in 25 fractions for the isolated primary HS of vertebral bodies is a viable option. Moreover, target therapies, particularly addressing the MAPK pathway, may change the future of the treatment for HS cases.

## Figures and Tables

**Figure 1 brainsci-12-00958-f001:**
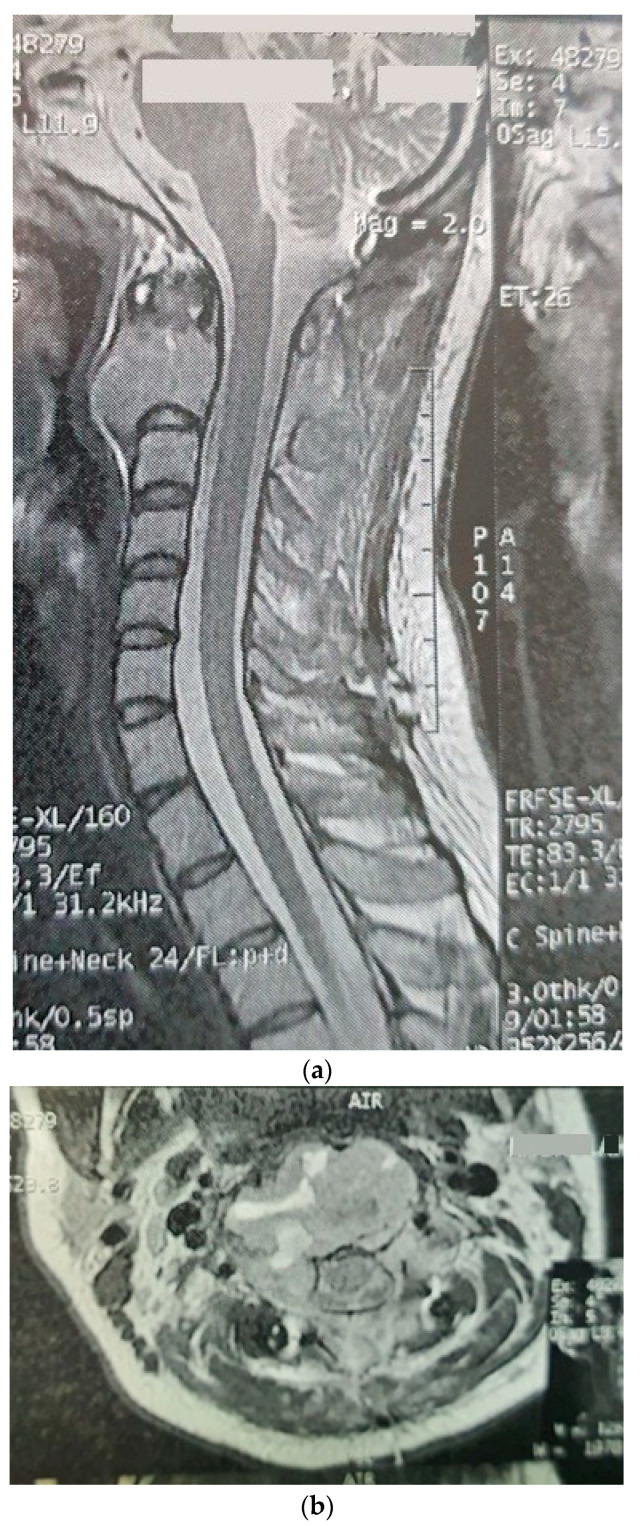
Sagitta (**a**) and axial (**b**) gadolinium-enhanced MRI showing the tumor in the C2 body.

**Figure 2 brainsci-12-00958-f002:**
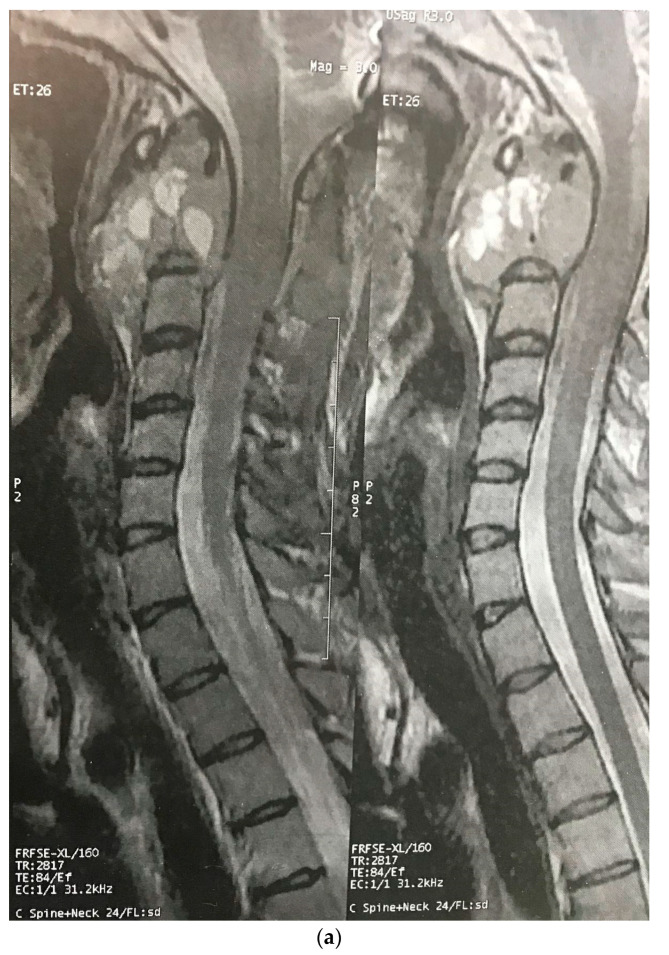
Sagittal (**a**) and axial (**b**) gadolinium-enhanced MRI after the progression of the tumor that caused severe dysphagia.

**Figure 3 brainsci-12-00958-f003:**
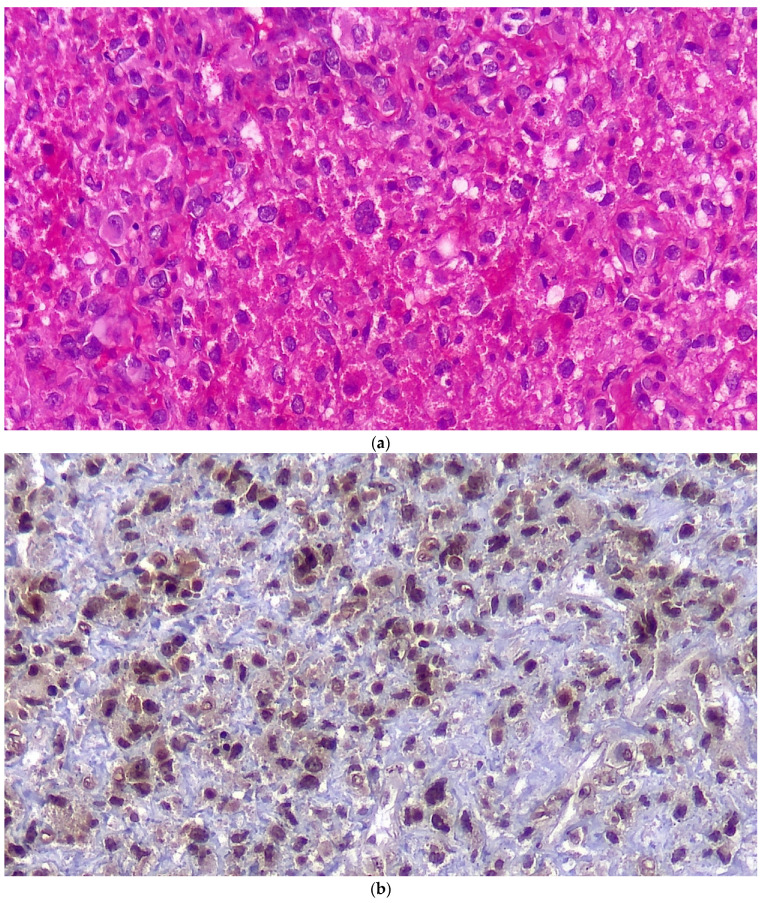
Histopathologic examination in regular H&E staining (**a**)**,** immunostaining for S100 (**b**) and for CD68 (**c**).

**Figure 4 brainsci-12-00958-f004:**
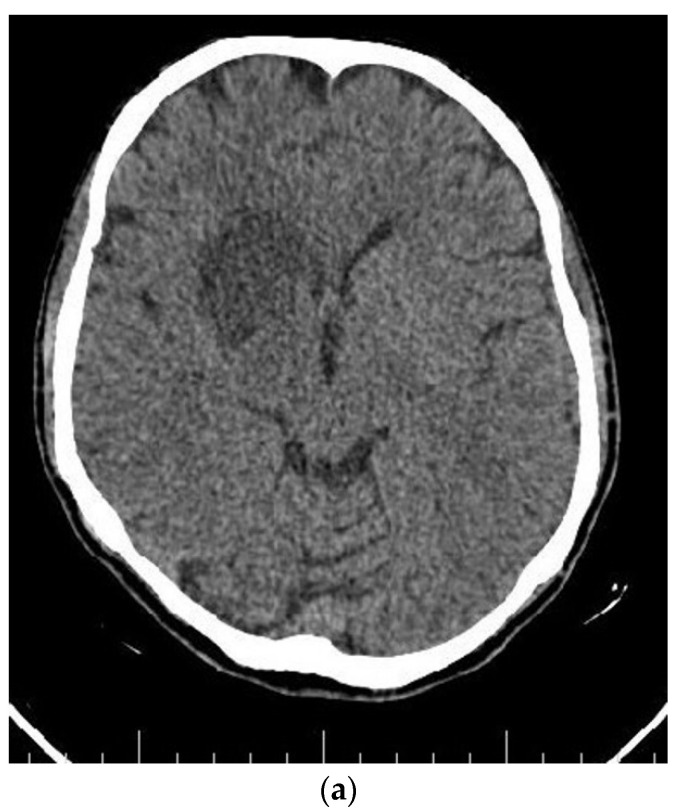
Axial (**a**) CT scan and MRI (**b**) and MR angiography (**c**) showing total occlusion of the right carotid artery along with a lesion in basal ganglia.

**Figure 5 brainsci-12-00958-f005:**
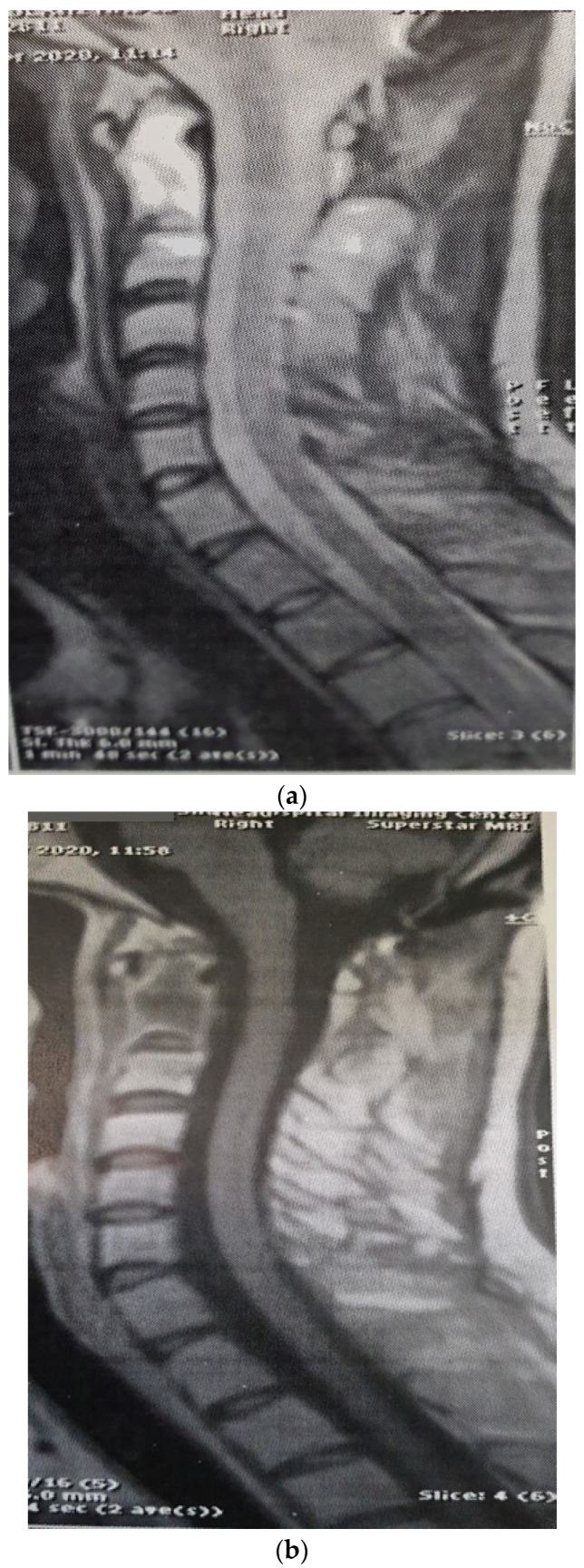
Post-radiotherapy MRI, sagittal T2W (**a**), sagittal T1W (**b**) showing a very good response to treatment.

**Figure 6 brainsci-12-00958-f006:**
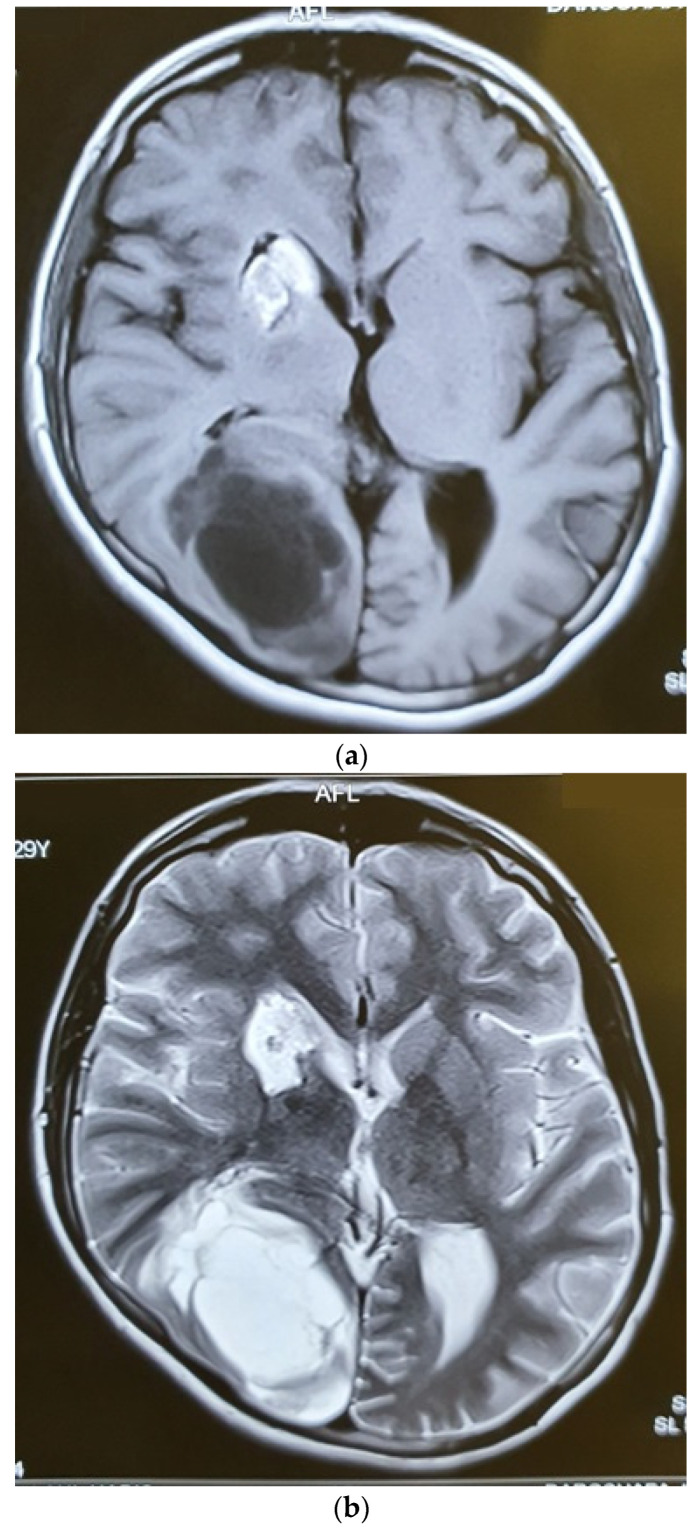
Axial T1W (**a**), T2W (**b**), and Gad-Enhanced MRI (**c**) showing the large cystic mass in the right parietooccipital area together with the old lesion in basal ganglia.

**Table 1 brainsci-12-00958-t001:** Summary of the main cases published in the literature, with description of the site, histological characteristics, treatment and outcome.

Author/Year	Age/Sex	Location	IHC	Treatment	Follow-Up
Pileri et al. (2002) [8]	26, male	T4, T5	CD 68, lysozyme, S100, CD45	Radiotherapy and autologous bone marrow transplantation at the time of relapse	After 3 years, alive in complete remisssion
Kaushal et al. (2012) [6]	17, male	L3	LCA, vimentin, CD68, CD163, Ki67 20%	Surgery + radiotherapy	Disease-free 3.5 years after surgery
Lin et al. (2012) [7]	27, female	L3	Cd68, vimentin, S100	Surgery (spondylectomy) without adjuvant treatment	No recurrence after 2 years
Patnaik et al. (2012) [2]	22, male	T8, T9	S100, CD38, Lysosyme	Partial excision without adjuvant treatment	Alive after 5 months
Buonocore et al. (2005) [5]	3, male	L4, T6	CD68, focal S100, CD15	Surgery and radiotherapy just to L4 and chemotherapy, and thereafter, chemoradiotherapy to pulmonary metastases	Two pulmonary nodules developed no change in the T6 lesion
Sohn et al. (2010) [9]	63, male	Multiple thoraco lumbar vertebra	CD68, S100 protein, CD31, CD99, vimentin weakly positive for: CD21, CD4, lysozyme	The patient refused all treatments	Died after 2 months
The current case	29, female	C2	CD45, CD43, CD68, S100	Surgery and radiotherapy	Brain metastasis after 9 months

## Data Availability

All data generated or analyzed in this study are included in the article.

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
