# Peer review of "Histiocytic Sarcoma Involving Cervical Vertebra: A Case Report and Review of the Literature"

_brainsci, 2022, doi:10.3390/brainsci12070958_

Round 1
Reviewer 1 Report
Dear Authors,
the paper presents an unusual spinal Histiocytic Sarcoma in a patient with history of cardiac "fibroma" with very bad outcome.
The HS was decently treated with radiotherapy that is indicated as a good option for this disease in such site.
Given that diagnosis of HS is pivotal in the paper histological data should be better described and histological figures (both HE and IHC) included.
Additionally, be aware that the name of the patient can be seen in the actual figures that should be anonymized!
Please find annotated file attached and some nodes below.
-----------
Introduction:
line 28: Histiocytes do not derive from mesenchymal cells (while it may be the case for other type of antigen presenting cells)
line 30: I don’t believe that “etc” is appropriate for a scientific report.
Line 35: Sentence to be rephrased
Line 39 substitute “which” with “and”
Case presentation
The pathological description is poor and deserves more efforts in the description of morphological features of the lesion (large, small cells? Epithelioid? Spindled? Haemophagocytoses which is unusual, was it diffuse or focal? Was there fibrosis? What S100 positive in all cells?)
Discussion
Please, see the file attached.
In general the histological description should be moved to the “Results” section where the description is very poor.
Line 142: please explain how the low proliferative index may underlie good response to therapy. Maybe it could identify a less aggressive HS.
From line 143 to line 173 it looks like a list of cases that partially replicate the table. It could be rephrased comparing age, clinical presentation and therapy all together.
Figures:
The patient’s name can be seen in many radiological images, please anonymize.

Author Response
DearEditor/Reviewer
Thank you for your very nice comments. We have considered these comments along with other reviewers’ comments and tried to fulfill what we were asked for. The manuscript has been changed accordingly.
Sincerely yours
Corresponding Author
Given that diagnosis of HS is pivotal in the paper histological data should be better described and histological figures (both HE and IHC) included. Additionally, be aware that the name of the patient can be seen in the actual figures that should be anonymized!
Thank you for your nice comments. We edited the manuscript regarding all your comments. Images are anonymized. Histological figures are included and other comments are also considered as follows.
-----------
Introduction:
line 28: Histiocytes do not derive from mesenchymal cells (while it may be the case for other type of antigen presenting cells)
line 30: I don’t believe that “etc” is appropriate for a scientific report.
Line 35: Sentence to be rephrased
Line 39 substitute “which” with “and”
Thank you for your attention. These errors are corrected in the manuscript
Case presentation
The pathological description is poor and deserves more efforts in the description of morphological features of the lesion (large, small cells? Epithelioid? Spindled? Haemophagocytoses which is unusual, was it diffuse or focal? Was there fibrosis? What S100 positive in all cells?)
Thank you for these comments. The mentioned details, which is really necessary, are added to the manuscript along with images of IHC.
Discussion
In general the histological description should be moved to the “Results” section where the description is very poor.
Histopathological description, is presented in the “case report” section. We added some details to it. Because this is a diagnostic challenge, pathological discussion is also included in the “discussion” part.
Line 142: please explain how the low proliferative index may underlie good response to therapy. Maybe it could identify a less aggressive HS.
Nice comment. We rephrased this part of the manuscript and added a description.
From line 143 to line 173 it looks like a list of cases that partially replicate the table. It could be rephrased comparing age, clinical presentation and therapy all together.
We have changed the manuscript addressing your nice comment. However, because of rarity of the disease and variation in case scenarios, it is difficult to stratify them in specific categories
Figures:
The patient’s name can be seen in many radiological images, please anonymize.
This is a perfect comment. We have edited the images accordingly

Reviewer 2 Report
The authors report a case of histiocytic sarcoma (HS) of the vertebra.
Introduction, line 36: HS is hypothesized to differentiate from any B cell lymphoma even small B cell lymphoma, not just diffuse large B cell lymphoma.
Figure 5 and figure 4 should be switched to keep them in order in the paper.
Figure 5. What is the explanation for the cystic mass in the right parietooccipital area?
How are the author certain of this diagnosis? HS is a diagnosis of exclusion. Was BRAF V600E mutation or IGH performed? IGH rearrangement is often positive in HS.
The authors also do not discuss new treatments such as MAPK targeted therapy.
Author Response
Response To Reviewer 2 comments
Thank you for your very nice comments. We have considered these comments along with other reviewers’ comments and tried to fulfill what we were asked for. The manuscript has been changed accordingly.
Comments and Suggestions for Authors
The authors report a case of histiocytic sarcoma (HS) of the vertebra.
Introduction, line 36: HS is hypothesized to differentiate from any B cell lymphoma even small B cell lymphoma, not just diffuse large B cell lymphoma.
Thank you for your comment, correction was made accordingly.
Figure 5 and figure 4 should be switched to keep them in order in the paper.
This is correct. I reversed their orders and added another image according to a comment made by another reviewer.
Figure 5. What is the explanation for the cystic mass in the right parietooccipital area?
The most probable diagnosis is a metastatic lesion. But, because the patient refused biopsy or surgery we do not have a definitive pathological diagnosis. This is defined in the manuscript.
How are the author certain of this diagnosis? HS is a diagnosis of exclusion. Was BRAF V600E mutation or IGH performed? IGH rearrangement is often positive in HS.
Diagnosis of HS is a challenge. A definitive diagnosis is difficult to make. Discussion about pathological diagnosis defines factors helping in diagnosis. The process of reaching the diagnosis is defined in the text and is concordant with the accepted approach in other reported cases. We did not assess BRAF V600E and IGH re-arrangement. We believe that these molecular markers are very useful for diagnosis, although they are neither sensitive nor specific enough. I added this discussion to the manuscript.
The authors also do not discuss new treatments such as MAPK targeted therapy.
This is a nice comment. Our patient has no access to NGS and target therapy, However, this is an interesting treatment option. We added some parts to our discussion and conclusion to cover this subject.

Round 2
Reviewer 1 Report
Dear Authors,
the manuscript has significantly improved.
Figures are ok.
Please find below some additional suggestions, some of which are relevant:
Line 69: S100 is mentioned to be negative in the debulking sample but following description and histological figures indicate that it is positive.
Line 77: Where the authors mention infiltration of "stroma" a brief description on how the bone tissue is involved (substituted or partially evident or anything else) since the lesion is in a vertebra could be appropriate.
Line 83: "IHC was positive" may be substituted with the more appropriate "By immunohistochemistry, atypical cells were positive for .... "
Lines 100, 250 and Table 1 : given that the patient is female, "he" and "male" should be corrected
Line 175: The sentence "HS shows the highest positivity of BRAF V600E among histiocytic and dendritic cell tumors" is not correct since Langerhans cell tumors and Erdheim Chester diseases are those with higher incidence of BRAF Val600Glu mutations. The data from reference [13] have been invalidated by multiple papers, I would suggest to say that "BRAF Val600Glu is a common mutation among histiocytic and dendritic cell tumors, including some HS cases [15]" .
Line 252: The sentence "antibodies that target different molecules in this pathway, such as BRAF, MAPK, KRAS, and MEK" is partially incorrect since, at present only 1 IHC antibody exist that is VE1 identifying BRAF Val600Glu mutated protein. Mutations on the other mentioned genes do not have a specific antibody. The only antibody that could help in the identification of MAPK pathway activation is phospho-ERK which is not gene specific.
Table1: for the current case some fields are empty and this is a pity since this impedes comparison with previously reported cases

Author Response
To Reviewer 1
Thank you so much for reviewing the current manuscript. There were some mistakes that were missed during our last edit. All of the mentioned points are reconsidered.
Line 69: S100 is mentioned to be negative in the debulking sample but following description and histological figures indicate that it is positive.
The first sample was through the pedicle has a smaller volume relative to the second operation. S100 negativity may be due to this fact that the biopsy was taken from an area of the tumor that was negative for s100. In line 74-84 it is mentioned that in the second operation by anterior approach, another sample was taken which was positive for S100. I added a point to the manuscript to bold this difference.
Line 77: Where the authors mention infiltration of "stroma" a brief description on how the bone tissue is involved (substituted or partially evident or anything else) since the lesion is in a vertebra could be appropriate.
It was mentioned in pathological description of the specimen, but was omitted. So I added this sentence to the manuscript: “Some of the particles were bordered by tiny bony trabeculae.”
Line 83: "IHC was positive" may be substituted with the more appropriate "By immunohistochemistry, atypical cells were positive for .... "
This phrase is edited in the manuscript.
Lines 100, 250 and Table 1 : given that the patient is female, "he" and "male" should be corrected
Thank you for pointing out these errors. They are all corrected in the manuscript.
Line 175: The sentence "HS shows the highest positivity of BRAF V600E among histiocytic and dendritic cell tumors" is not correct since Langerhans cell tumors and Erdheim Chester diseases are those with higher incidence of BRAF Val600Glu mutations. The data from reference [13] have been invalidated by multiple papers, I would suggest to say that "BRAF Val600Glu is a common mutation among histiocytic and dendritic cell tumors, including some HS cases [15]" .
Thank you for your nice comment. I have edited the text as you mentioned above.
Line 252: The sentence "antibodies that target different molecules in this pathway, such as BRAF, MAPK, KRAS, and MEK" is partially incorrect since, at present only 1 IHC antibody exist that is VE1 identifying BRAF Val600Glu mutated protein. Mutations on the other mentioned genes do not have a specific antibody. The only antibody that could help in the identification of MAPK pathway activation is phospho-ERK which is not gene specific.
I meant “potential” antibodies against different parts of this pathway. However, to eliminate any confusion, this part of the manuscript is edited.
Table1: for the current case some fields are empty and this is a pity since this impedes comparison with previously reported cases
This is totally true. Some parts in the table were missed. They are edited now.

Reviewer 2 Report
The authors have tried to address the reviewer’s concerns. The authors rely on only morphology and immunohistochemistry with exclusion as much as is possible from other entities.
The authors did not address newer molecular tests of immunohistochemistry such a BRAF stains or BRAF mutation studies. They also did not perform IGH rearrangement studies. It was not clear in the comment if these tests were not available or just not performed.
Author Response
to Reviewer 2
Thank you for your comment. Unfortunately, these tests are not available in our country. For some patients, samples are sent to other countries to perform the required test. However, this is time-taking and expensive and can not be suggested for all patients.
